# At-Hook Motif Nuclear Localised Protein 18 as a Novel Modulator of Root System Architecture

**DOI:** 10.3390/ijms21051886

**Published:** 2020-03-10

**Authors:** Marek Širl, Tereza Šnajdrová, Dolores Gutiérrez-Alanís, Joseph G. Dubrovsky, Jean Phillipe Vielle-Calzada, Ivan Kulich, Aleš Soukup

**Affiliations:** 1Department of Experimental Biology of Plants, Faculty of Science, Charles University, 128 44 Prague, Czech Republic; 2Departamento de Biología Molecular de Plantas, Instituto de Biotecnología, Universidad Nacional Autónoma de México, Avenida Universidad 2001, Colonia Chamilpa, Cuernavaca, Morelos 62210, Mexico; 3Laboratorio Nacional de Genómica para la Biodiversidad, Centro de Investigación y de Estudios Avanzados del Instituto Politécnico Nacional Irapuato, Guanajuato 36821, Mexico

**Keywords:** AT-hook motif nuclear protein 18, AHL18, At3G60870, *Arabidopsis*, Lateral root development, Root apical meristem, Cell proliferation

## Abstract

The *At-Hook Motif Nuclear* Localized *Protein* (*AHL*) gene family encodes embryophyte-specific nuclear proteins with DNA binding activity. They modulate gene expression and affect various developmental processes in plants. We identify *AHL18* (At3G60870) as a developmental modulator of root system architecture and growth. *AHL18* is involved in regulation of the length of the proliferation domain and number of dividing cells in the root apical meristem and thereby, cell production. Both primary root growth and lateral root development respond according to *AHL18* transcription level. The *ahl18* knock-out plants show reduced root systems due to a shorter primary root and a lower number of lateral roots. This change results from a higher number of arrested and non-developing lateral root primordia (LRP) rather than from a decreased LRP initiation. The over-expression of *AHL18* results in a more extensive root system, longer primary roots, and increased density of lateral root initiation events. *AHL18* is thus involved in the formation of lateral roots at both LRP initiation and their later development. We conclude that *AHL18* participates in modulation of root system architecture through regulation of root apical meristem activity, lateral root initiation and emergence; these correspond well with expression pattern of *AHL18*.

## 1. Introduction

The root system is essential for plant anchorage, water and nutrition acquisition, and mediating interactions with the soil environment and microorganisms. It is a dynamic structure, modulating its growth according to the requirements of the whole plant. Lateral root (LR) initiation and function of individual apical meristems significantly contribute to the establishment of root system architecture in heterogeneous soil environments, and are under precise developmental regulation. The formation of LR primordia (LRP) [1] is a complex sequential process [2,3], which in conjunction with neighboring tissues [4,5,6], starts in the parent root pericycle in seed plants. Auxin signaling was identified as a critical component of the process [7,8,9,10] alongside other phytohormones, transcription factors, and epigenetic control of the regulatory networks [11]. The regulatory network modules, driving LRP development [3,12], might act as switches allowing the LRP to gradually develop into LRs. In search for new regulatory factors involved in this complex interaction, we perform forward genetic screening for gene-trap lines with an expression pattern associated with LR development. The target gene of one of these lines was identified as AT-HOOK MOTIF CONTAINING NUCLEAR LOCALIZED PROTEIN 18—*AHL18* (At3G60870).

The *AHL* family represents common genes specific to terrestrial plants [13]. In *Arabidopsis thaliana* there are 29 paralogs divided into two clades according to presence of intron and number of AT-hook motifs [14]. The AHL proteins contain a unique combination of one or two AT-hook domains, allowing them to bind to the minor groove of DNA in the specific AT rich sites and in Plants and prokaryotes conserved domain (PPC), which is responsible for their nuclear localization and interaction with other proteins. They can either, form homo- or hetero-oligomers with other AHLs or interact with some transcription factors [13,15]. They are also known to modulate chromatin structure and regulate gene expression at an epigenetic level [16].

AHLs take part in various developmental processes such as negative leaf senescence regulation (AtAHL27, [17]); reproductive organs patterning and differentiation downstream of AGAMOUS (AtAHL21, [16]); flowering initiation [18]; plant defense and immunity (AtAHL27, AtAHL20, AtAHL19, SolycAHL5, SolycAHL9, [19,20,21]); suppression of hypocotyl elongation in light (AHL29 and AHL27, [22]); modulation of GA biosynthesis (AtAHL15, [23]); redundant regulation of auxin biosynthesis (AtAHL 29, [24]); ABA mediated stress growth regulation [15]. The only currently known function of AHLs in roots is the cell non-autonomous interaction of AtAHL3 and AtAHL4 which is followed by definition of phloem and xylem boundaries within the developing root pro-cambium [25].

An intriguing property of AHL activity is the multilevel mode of their action (epigenetics, interaction with a transcription factor, simultaneous function as an activator and repressor of transcription) frequently bridging various regulatory pathways [15,16]. This modus operandi makes them rather potent and complex regulators, with localization of expression, as well as interactions between different AHLs playing a crucial role. The number of AHLs in *A. thaliana* and their overlapping expression domains might be explained by the frequently reported low phenotypic responses of single AHL mutants, most likely due to functional redundancy [13,22,24] and feedback regulation [15]. On the other hand, there is a rather strong phenotype of ectopic expression under constitutive promoter [26], which indicates high potential for non-specific regulation out of a narrow expression domain.

In the present study, we analyze the role of *AHL18* in the formation of early root system architecture of *A. thaliana* alongside its expression domain and protein localization. There is clear regulatory function of *AHl18* in the root apical meristem activity, onset of differentiation, LRP initiation and their latter development.

## 2. Results

### 2.1. Identification of AHL18 as a Gene Involved in Root Development and Analysis of its Expression

A collection of ~2000 enhancer and gene trap lines harboring Ds elements with β-glucuronidase (GUS) reporter gene [27,28] was screened for GUS expression in the pericycle and early LRP stages. Analysis was carried out on cleared roots under a microscope equipped with Nomarski optics. Line MGT180 was selected because its GUS expression was stable and reproducible in pericycle cells, and it was detectable during early stages of LRP formation and did not change with plant age. The T-DNA flanking sequence was determined using TAIL PCR [29] and a candidate gene involved in this expression pattern was identified as *AT-HOOK MOTIF NUCLEAR LOCALIZED PROTEIN 18* (*AHL18*, At3g60870). To validate this conclusion, we created *A. thaliana* lines *pAHL18::mCherry:mCherry* and *pAHL18::GUS.* High *AHL18* promoter activity was detected in the primary root apical meristem (RAM), differentiating xylem and an adjacent sector of the pericycle up to the root differentiation zone, in the early LRPs, and in the RAM of emerging and developed LRs (Figure 1).

Transcription of *AHL18* was also high in the innermost layer of lateral root cap and protoderm up to the transition zone. *AHL18* promoter activity was not found in the shoot, which was consistent with publicly available RNA seq data (https://apps.araport.org/thalemine/report.do?id=1017628). An overall similar pattern of expression was found when analyzing a translation fusion *pAHL18::AHL18:mRuby* line (Figure 2). Outside the RAM, the fusion protein was detected exclusively in xylem-adjacent sector of the pericycle (Figure 2C). Interestingly, there was no detectable expression in the RAM quiescence center and in the directly adjacent initial cells or the two to three cells above them (Figure 2D). *AHL18* promoter activity decreases through the elongation zone and disappears with the beginning of differentiation, except in the xylem-adjacent pericycle (Figure 1F,G and Figure 2E) and in the early stages of developing LRP (Figure 1F and Figure 2E). With the onset of periclinal divisions in LRP (occurring at about Stage 2 according to [9]) the promoter activity disappears. The transcription of *AHL18* reappears in the LRPs later, just prior LR emergence from the parent root, and maintains in growing LRs. *AHL18* expression in the RAM of LRs is similar to that in the primary root (Figure 2B). AHL18 protein localizes to the nucleus (Figure 2) in a pattern identical to promoter activity.

### 2.2. Modulation of AHL18 Expression Affects Root System Architecture

The *ahl18* knock out mutant showed a decreased total length of all roots within the root system (Figure 3A). In part, this was related to a decrease in the primary root length, though the difference was only 10.5% compared to the control WT plants (Figure 3B). The decrease in the total length of LRs was more pronounced and in the mutant, and was reduced by 48.2% of WT (ANOVA, *p* < 0.05, *n* = 64), due to a decreased number of LRs (Figure 3E).

The *AHL18* overexpression under a constitutive 35S promoter resulted in phenotype, which was in line with the observations described over. The overall length of the root system was increased in *rdr6* but not in Col-0 background. Primary and lateral roots were longer and number of LRs was higher (Figure 3C,E). This effect was visible starting from 5 DAG and significantly modified morphometric parameters of the root system in *p35S::AHL18* line and resulted in a more extensive theoretical cover of the rhizosphere (Figure 4). Over-expression lines also showed different angles of LR branching (Appendix A), which was not evaluated further in this study. The described over-expression phenotype was observed only in *rdr6* background, with reduced posttranscriptional silencing [30] and not in the Col-0 background with confirmed insertion of T-DNA. This response is in accordance with the quantity of mRNA of AHL18 as evaluated through q-PCR (Appendix A) and indicates posttranscriptional regulation of the 35S::*AHL18* transcript in Col-0. Phenotype of *35S::AHL18* overexpression lines also showed changes within the shoot. Leaves were curved and mis-shapen, inflorescences grew longer with procumbent tendencies (Appendix A).

In further search for the reason behind the lower number of LRs in root system in the *ahl18 knock out* mutant, we calculated the total number of LR initiation events along roots of the same age and compared these to the number of LRs in the branching zone [31]. The overall number of initiation events was slightly decreased in the *ahl18* mutant compared to WT. The main difference in the number of LRs is related to the lower percentage of those which developed as emerged LRs. Out of total LR initiations in *ahl18*, 25% remained unemerged and could be observed as arrested in various stages of development in the branching zone. In control WT plants, the percentage of arrested primordia was 11% (ANOVA, *p* < 0.05, *n* = 27). On the other hand, an increase of *AHL18* expression under the *35S* promoter significantly increased the number of developing LRs and the density of all LRP events (See Section 2.4.) Modulation of *AHL18* transcription thus affects LRP initiation under overexpression, but not in *ahl18*. This suggests that changes in the root system architecture in *ahl18* are related to a delay in the developmental transition from LRP to LR.

### 2.3. AHL18 Is Required for the Root Apical Meristem Activity

The data presented above show that *AHL18* modulates growth rates of the primary, as well as the lateral roots. In order to understand the source of differences in root growth, we further tested the RAM length of the primary root, which if compared with the length of fully elongated cortical cells should reflect cell production of RAM [32]. The length of the RAM proliferation domain was measured as a distance from the quiescence center (QC) to the beginning of the transition domain, represented by a cluster of the first three elongated cells of the cortex detected on both sides of the median longitudinal optical section of the root. The length of differentiated cortical cells was measured at the level of the first emerged LR with distinct meristematic and elongation zones (Figure 5).

There was no significant difference in the length of fully elongated cortical cells in the *ahl18* mutant compared to the WT, which is valid also for overexpression lines compared to the *rdr6* control. The significantly shorter meristematic zone in the *ahl18* mutant suggests that the *AHL18* is a positive regulator of cell production in the primary root. In line with this, plants overexpressing *AHL18* had longer RAM meristematic zones with shorter uniformly sized cells (as measured in the pericycle) (Figure 5G) than the respective control (*rdr6*), indicating their enhanced proliferation. This observation was further supported by the increased number of S-phase cell nuclei labelled by EdU copying the expression level of *AHL18* (Figure 6). Genotype *ahl18* had a significantly lower number of S-phase nuclei than respective wild type by 20.4%, while *35S::AHL18* plants had this number further increased by 37.7% (Figure 6A) suggesting differences in proliferation activities of the compared genotypes.

Therefore, we conclude that there was an obvious link between length of meristematic zone, cell proliferation activity, and level of *AHL18* expression. Thus, *AHL18* is involved in modulating the rate of cell production in the RAM thereby affecting root growth and root system architecture.

### 2.4. Role of AHL18 in Later Development of Lateral Root Primordia

Numbers of LRPs in the branching zone of the root were used to estimate the frequency of delayed or arrested LRPs [33] among emerged LRs, which were established in acropetal sequence. Their quantity responded consistently to modulated expression of *AHL18* (Figure 7). In the knock-out mutant *ahl18*, reduced numbers of LRs were brought about by a higher frequency of arrested or delayed LRPs (Figure 7B). These LRP were located in the branching zone. The LRPs in the unbranched zone of primary root were considered as developing, irrespective of their developmental stage and neighboring LRPs. This rough estimate was used for the purpose of a quantitative evaluation of the phenotype under gene transcription modulation.

The *ahl18* mutant produced a significantly higher proportion of arrested or delayed LRPs and a lower number of emerged LRs. The total number of LR initiation events was lower compared to the WT (Figure 7A). However, a shorter primary root and lower number of total LR initiation events maintained the same respective density of LR initiation events compared to the WT (Figure 8A). The increased density of arrested LRPs in the branching zone of primary root of *ahl18* (0.39 mm^−1^) comparing to the WT (0.16 mm^−1^) clearly indicates their greater abundance (Figure 8C). In the *ahl18* mutant, the arrested or delayed LRPs within the branching zone made up 50.7% of all LR initiation events while the same parameter in the respective WT was only 25.2%.

*AHL18* over-expression (*rdr6* background) had a significantly higher number of LR initiation events (Figure 7C), which also affected their density (Figure 8D,E). This increased the total number of arrested primordia (Figure 7D) as well as their density. However, the fraction of arrested or delayed LRPs present only in branching zone is significantly lower than in the respective wildtype. In the increased LR initiation background of *35S::AHL18*, the percentage of arrested or delayed LRPs was significantly decreased (ANOVA, *p* < 0,05, *n* = 22). Interestingly, the density of arrested or delayed LRPs in the branching zone was the same as in control (Figure 8F) and only 26.1% LRPs remained un-emerged in the branching zone (as compared to 18.4% in the respective control).

This analysis shows that the level of *AHL18* expression modulates the extent of LR initiation. Accordingly, in *35S::AHL18* roots, closely spaced LRPs were found (Figure 8G.). *AHL18* takes part in the regulation of LRP development during LR emergence and post-emergence growth. This is in agreement with observed promoter activity. *AHL18* transcription was detected in pericycle founder cells and Stage I LRPs but disappeared after stage II and was found again in the LRPs of stage VII and later (Figure 1). In most of the arrested or delayed LRPs within the branching zone, the *AHL18* promoter activity was not detected (Figure 7F).

### 2.5. AHL18 Related Phenotypes Are not Directly Affected by Auxin

The *AHL18* effect on LRP initiation events and modified numbers of LRs lead us to test whether AHL18-related phenotypes changes were directly connected with auxin concentration. Cultivation on medium supplemented with 1-naphthaleneacetic acid (NAA) induced the expected reduction in primary root growth and rapid LRP induction across all tested genotypes. The relative response of *ahl18* knockout mutants, *35S::AHL18* plants, and their respective controls remained without significant differences among the genotypes (Appendix A). Partial phenotype restoration was observed in the development of arrested or delayed LRPs in *ahl18* plants (Figure 9B). In seedlings treated with NAA, abundance of these LRPs in the branching zone decreased in *ahl18* plants to the level of WT plants (Figure 9A). Similar response of primary root growth and LR formation to exogenous auxin treatment in lines with loss-of- and gain-of-function of *AHL18* suggested no obvious changes in auxin sensitivity. The partial recovery of the number of arrested or delayed LRPs in *ahl18* could be related to auxin metabolism downstream of *AHL18* action.

## 3. Discussion

### 3.1. Role of AHLs in Regulation of Lateral Root Development.

We identified At-hook motif nuclear localized protein AHL18 as a novel regulator of root system development and architecture. The modulation of *AHL18* expression affected primary root growth, as well as the development of LRs—crucial processes shaping the spatial configuration of the root system.

There are 29 paralogues of the AHL proteins in *Arabidopsis* with conserved structure and predicted redundant behavior [13]. They are predicted to form homo and hetero oligomers (trimers) via the plants and prokaryotes conserved (PPC) domain [14,15,25,34]. The current knowledge indicates, that *AHL* genes are able to bridge regulation of several signal pathways acting as a nodes of the regulatory network integrating different inputs [15,35] The significance of overlapping expression domains, possible specific interactions among individual AHLs, and the resulting selectivity of particular oligomers for DNA binding, as well as interaction with other proteins via the PPC domain, could further extend and specify the mode of AHL action. Their role as expression modulators might influence various levels of the process. Due to ability of AHLs to interact with conserved scaffold/matrix attachment regions (MARs), direct remodeling of chromatin might suppress transcription for large stretches of DNA [14]. Rice homolog OsAHL1 thus directly binds to the promotor region and modulates expression of several genes for drought resistance [36]. ESCAROLA (ESC)/AHL27 and SUPPRESSOR OF PHYTOCHROME B-4 #3 (SOB3)/AHL29 redundantly regulates auxin biosynthesis during hypocotyl light response binding to an MARs region of the YUCCA 9 (YUC9) promoter and suppressing its activity also via SWI2/SNF2-RELATED 1 (SWR1) histone modifying complex [24]. Additional modes of regulation by AHLs may relate to their ability to bind transcription factors via specific sequence of PPC domain and thus modulate activity of target genes [13,37]. The complexity of AHL action is further enhanced by feedback loop and phosphorylation controlled interactions with other modulators of gene expression. This was recently reported for AHL10 [15], a plant growth modulator under low-water potential stress, affecting jasmonic acid and auxin-synthesis and controlling developmental regulators such as SHOOT MERISTEMLESS (STM) [15].

Due to the absence of an AHL18 probe in chip experiments, there is limited information on expression data of the gene. However, the sequencing outputs indicate, in accord with our presented results, that AHL18 is a regulator of root development, whose activity is located at the meristematic and differentiation zones of the primary and lateral roots, with extension to the xylem sectors of pericycle, which is competent to give rise to LR founder cells. There is a consistent trend in changes of root system parameters across *ahl18* (knock-out insertion mutant), AHL18 wild type, and overexpression (35S constitutive promoter) lines, which are modified in AHL18-dependent manner.

### 3.2. Root Apical Meristem and AHL18

The transcription of *AHL18* is restricted to the developing tissues of the root tip: the protoderm within the meristematic and elongation zones and the two outermost layers of the xylem sector of the central cylinder (protoxylem and pericycle). It is excluded from the quiescent center and adjacent initials and neighboring cells, a sub-domain defining the stem cell niche and stem cells via complex hormonal, epigenetic, transcriptional and mechanical regulatory network [38,39,40]. AHL18, thus, seems to be functionally linked, not with initials, but with their derivatives (daughter cells), particularly in the xylem sector of the procambium and corresponds to AHP6 expression pattern, which is an early marker of this sector [41]. Transcription of *AHL18* is detectable also in the protoderm of proliferation and transition domains of the RAM. It is rather interesting in this context that *AHL18* activity can be found in root cap-protoderm initial cells and its derivatives, but not in the columella initial cells. The cell autonomous effect should be expected from identical localization of both promotor activity and protein presence.

The functional role of *AHL18* is indicated by changes in meristematic zone length, which increases according to level of *AHL18* transcription. An increase in the RAM length in *AHL18* overexpression lines is apparently caused by a higher cell production, which is reflected in number of cells which passed the mitotic S-phase during 45-min EdU exposure period. As no significant difference in fully elongated cortical cell length was found, we can conclude that an increased root length resulted from increased cell production. Significantly smaller cells of longer meristematic zone in *AHL18* over-expression lines indicate that the pace of cell division and their volume of growth are not maintained. The meristematic activity and cell size are under developmental and environmental control [42,43], but regulatory feedback balances cell size, growth, and division rate [43], at least in the shoot apical meristem. The length of the meristematic zone also depends on cell cycle duration and rate of transition of meristematic cells into the elongation phase [44,45,46]. The transition domain presents a developmental boundary between dividing and differentiating cells. Its position is controlled by regulatory balance of auxin and cytokinine [45], which positions the auxin minimum [47] initiating the cell elongation (i.e., the differentiation). The regulation of auxin accumulation, and its degradation through *ARR1* signaling, is dependent on *PIN5* and *GH3.17* genes [48]. A shortened meristematic zone with rapid transition from basal portion of proliferation domain into transition domain is one of developmental possibilities that lead to shorter primary root in *ahl18*. However, a shorter RAM proliferation domain can be also accompanied by a transition to elongation which is slower than in the WT, as evidenced by the estimation of the number of cells that transit to elongation in *Arabidopsis homolog of trithorax1* (*atx1*) mutants [49]. Members of *PLETHORA* family control the transition to elongation/differentiation in a dose-dependent manner with feedback to/from polar auxin transport [50]. The balance of proliferation and elongation is further modulated through the balance of reactive oxygen species (ROS) via UPBEAT1 (UPB1) [51]. The ROS are tightly connected with ABA-mediated stress response and their higher content regulate root meristem size and activity by directly controlling *PLETHORA* transcription and auxin accumulation in the root tip [52]. The distribution of ROS is another important factor which controls the transition between cell proliferation and differentiation with role in stem cell niche maintenance in *Arabidopsis* roots, with increased quiescent center cell division and distal stem cell differentiation in lowered ROS levels and in auxin independent pathway [53]. The molecular engagement of AHL18 in the regulatory network of the RAM requires further elucidation.

### 3.3. LR Initiation and Development

The formation of new LRs is a multistep process confined to the competent sector of the pericycle [54,55,56]. These cells exhibit a specific and prolonged cell cycling pattern [55,57,58] retaining their meristematic character (cell size, vacuolization, large nuclei and dense cytoplasm, G2 phase, cell division) longer than other root tissues [55,59,60]. ABERRANT LATERAL ROOT FORMATION 4 (ALF4) [61] is modulator of lateral root initiation maintaining the pericycle in a mitotically competent state [62].

The presence of AHL18 in in the competent sector of the pericycle might anticipate its role in the process. The absence of phenotype in this aspect for *ahl18* mutant might be attributed to gene redundancy within the family. Most of the single mutants of AHLs described at this point do not have obvious phenotype [26,34]. However, the over-expression of *AHL18* increases the total number of LR initiation events and disturbs correct spacing of initiated LRP. Ectopic presence of AHL18 should be interpreted carefully until its regulatory role in LR initiation is fully established but there is a known connection between pericycle cell length and LRP spacing [55,63]. However, in *AHL18* over-expression lines, the pericycle cell length was modified only in the RAM, but not in the differentiation zone.

*AHL18* is present not only in the cells of the competent pericycle sector, where the initiation takes place, but takes part also during LRPs later development. The AHL18 disappears from the tissues of LRP during stage I and is not transcribed until the stage VI–VII. According to the biphasic model of LRP development [64], the AHL18 presence is associated with later LRP developmental stage. The regulatory events, required to build an active meristem of an LRP from pericycle founder cells, with a completely autonomous stem cell niche, recapitulates the development of a primary root tip. The coordination of patterning in the new meristem depends on the PLETHORA transcription factors [65], and de novo formation of the QC [64] takes place as early as from stage IV–V, when LRP passes across overlaying endodermis, gaining its radial symmetry and auxin signaling autonomy from the parent root. Formation of a functional stem cell niche in the developing LRP [64,66,67] is related to the SHORT ROOT (SHR)–SCARECROW (SCR) module [68]. Various molecular markers indicate that a functional stem cell niche is present before the LRP emerges [64]. However, the time of emergence seems to be the major activation switch of LR meristem [61]. During this developmental period, AHL18 reappears in the central domain of the two outermost layers of LRP, except for central most position above the newly formed QC, similar to its localization in the primary root tip.

Development of an active meristem is associated with outgrowth of new LRs out of the parent root. This step is connected with extensive changes in the mechanics of the tissues of both LRP and the parent root [68]. Elongation of cells in the basal part of the central domain of LRP changes hydraulic properties [69]. Symplastic connections with surrounding tissues [4], cell wall remodeling, and separation of cells from overlaying tissues [70] are all part of this process. However, not all of the initiated primordia develop into emerged lateral roots, and fraction of the arrested or delayed and slowly developing LRP remain among the developed laterals [71]. In *ahl18* plants, this population is significantly higher. The regulatory mechanisms behind this developmental retardation are not completely clear and we believe that further disentangling of regulatory network of AHL18 might fill this gap. The role of AHL18 in meristem activity and cell division promoting function in primary RAM are highly consistent with this role. The high frequency of arrested or delayed LRPs in *ahl18* can be partially compensated by external auxin treatment, indicating that in this particular process, AHL18 acts upstream of auxin signaling.

AHL18 is as a novel regulatory element taking part in early stages of development of root system architecture. Its activity in the regulatory network controlling the root apical meristem proliferative activity, and onset of differentiation modulates growth of primary root. The experimental changes of *AHL18* expression affect the initiation of lateral root primordia and also their latter development. Increase in number of arrested/developmentally delayed primordia among already emerged lateral roots is one of traits associated with *ahl18* phenotype. The presence of arrested primordia is well known phenomenon, which still awaits clarification of the underlying regulatory network. We believe that a deeper understanding of AHL18 regulatory function will extend our understanding of the processes mentioned above.

## 4. Materials and Methods

### 4.1. Plant Material and Growth Conditions

An *A. thaliana* (L.) Heynh. insertion mutant line for *AHL18* gene (At3g60870, Sail_346_C06) with T-DNA flanking sequence insertion in exon at position 469 bp after the start codon, was obtained from publicly available collections (NASC, Nottingham Arabidopsis Stock Centre, Loughborough LE125RD, United Kingdoms) and genotyped for the presence of the T-DNA insertion using pre-designed primers (iSect Primers tool, signal.salk.edu) (Appendix A). The wild-type line for *AHL18* was selected from *ahl18* heterozygous progeny.

To monitor plant growth, seeds were surface sterilized and plated on 0.2× MS medium supplemented with 1% sucrose and 0.8% agar. For exogenous auxin application, 1-naphthaleneacetic acid (NAA) was added into MS medium to a final concentration of 5 nM, 50 nM, and 100 nM (after autoclaving). After a 3-day vernalisation period (4 °C, in dark), Petri dishes were transferred to the cultivation room (21/18 °C, 16/8 h day/night, 60% humidity). The Petri dish holder maintained the dishes in a position 45° to the horizon. Plant growth was recorded at 5, 7, and 9 days after germination (DAG). The root system grown on the agar surface was analyzed after scanning of dishes at high resolution (1200 dpi, 48-bit) and image analysis was performed using SmartRoot software [72] The length of primary and lateral roots, as well as their numbers were recorded.

### 4.2. Molecular Cloning and Transgenic Lines

Restriction site and Gateway^®^ system cloning technologies were combined to generate transcriptional (*pAHL18::GUS, pAHl18::mCherry:mCherry*) and translational (*pAHL18::AHL18:mRUBY, 35S::AHL18:GFP*) reporter lines using in silico design software Geneious 10.0.9. The complete coding sequence and the promoter region (1.8 kb upstream) were amplified from Col-0 gDNA, which was isolated from leaves of *A. thaliana* (Ecotype Col-0) by CTAB method [73]. Fragments were recombined into entry vectors and subsequently into destination vectors (for cloning details see Appendix A). Transgenic lines were generated by the floral-dip method using *Agrobacterium tumefaciens* strain GV3101 in *Arabidopsis* Col-0 background. *AHL18* overexpression lines (*35S::AHL18:GFP*) were cloned to Col-0 and *rdr6* background, deficient in siRNA silencing to avoid post-transcriptional silencing (co-suppression) of the transgene [30]. At least three independently transformed lines were selected and evaluated.

### 4.3. qPCR

The level of transcription was quantified by reverse transcription q-PCR. RNA was isolated from 30 mg of root system biomass using the Monarch Total RNA Miniprep Kit (NEB #T2010) and reverse transcribed with TaqMan™ Reverse Transcription Reagents. LightCycler 480 (Roche, Basel CH-4070, Switzerland) and Generi Biotech SG PCR Master Mix with primers (Appendix A). Technical triplicates were used and specificity of the PCR was verified by melting curve analysis (using the LightCycler 480 software). The PCR efficiency for each amplicon and the Cq values for each sample were calculated using the software LinRegPCR 2015.3 [74]. The calculated concentrations were normalized to the expression of the internal expression standard *EF1α.*

### 4.4. Sample Processing and Microscopy

Confocal images of *pAHL18::AHL18:mRUBY* and *pAHL18::mCherry:mCherry* lines were recorded by confocal Zeiss LSM 880 or spinning disk Yokogawa W1 microscope on unstained live plants. For *GUS* assay, *pAHl18::GUS* plants (7 DAG) were fixed in 90% *v/v* acetone (−20 °C, 30 min) and stained in 0.5 mM K_4_Fe(CN)_6_, 0.5 mM K_3_Fe(CN)_6_, 100 mM phosphate buffer (pH 7.0), and 0.19 mM 5-bromo-4-chloro-3-indolyl-β-D-glucuronic acid (X-gluc) for 1 h at 37 °C. After staining, plants were gradually transferred into 65% glycerol and mounted in NaI-based clearing solution to obtain whole-mount root preparations [75]. Root anatomy, the number of root primordia and their GUS activity were documented with Nikon Eclipse-90i microscope equipped with DIC optics and Andor Zyla 5.5 (Oxford Instruments, Abingdon OX13 5QX, England) or Nikon Ds-Fi3 (Nikon Instruments Inc., 1076 ER Amsterdam, Netherland) cameras. The primordia were classified into two categories: (i) Developing LRPs located distally to the youngest fully established LR (LRPs in the LR formation zone in accordance to classification described [31]; and (ii) LRPs located proximally to the youngest emerged LR (those found within the branching zone). The latter ones were considered to be arrested or delayed in the emergence and formation of LR [33].

### 4.5. EdU Labelling

A 5-ethynyl-2-deoxyuridine (EdU)-based assay for S-phase detection was performed according [76] using EdU-Click 488 (Baseclick GmbH, 82061 Neuried, Germany). 7 DAG seedlings were cultivated in 20 μM EdU solution for 45 min at 25 °C, fixed in 4% formaldehyde (30 min at room temperature), and washed 3 times, 15 min each, in PBS. Fluorescent labelling of EdU was set for 45 min at room temperature in the dark, followed by three washes in PBS. The number of nuclei passing through the S-phase of mitosis was estimated from 20 standardized optical sections per root (Z distance—1μm) in FIJI ImageJ. Threshold segmented nuclei were counted by 3D object counter to avoid repetitive registration.

### 4.6. Statistics

Macroscopic phenotype evaluation was performed in three independent experiments, exogenous auxin application was repeated twice. Plants from each repetition were then collected for further microscopy. Total number of replicate plants per line are indicated with associated graphs and within the text of results. Statistical analysis was performed using NCSS 9.0.15 software [77]. One-way and nested ANOVA with multiple comparisons Tukey-Kramer tests or Two-sample *t*-test were used to analyze differences among genotypes. No data transformation was necessary.

## Figures and Tables

**Figure 1 ijms-21-01886-f001:**
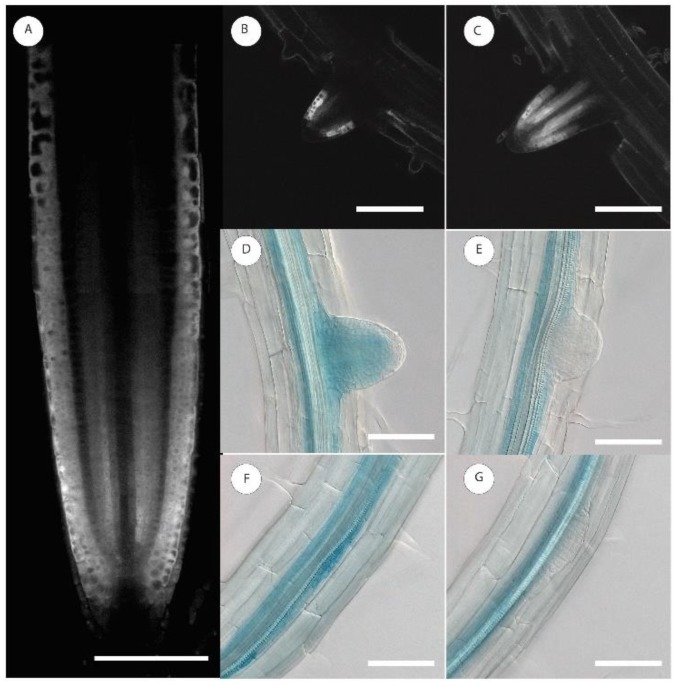
Localisation of *AHL18* promotor activity in the primary root apical meristem and during lateral root primordium development. (**A**–**C**) *pAHL18::mCherry:mCherry* (**D**–**G**) *pAHL18::GUS*. Scale bars = 100 µm.

**Figure 2 ijms-21-01886-f002:**
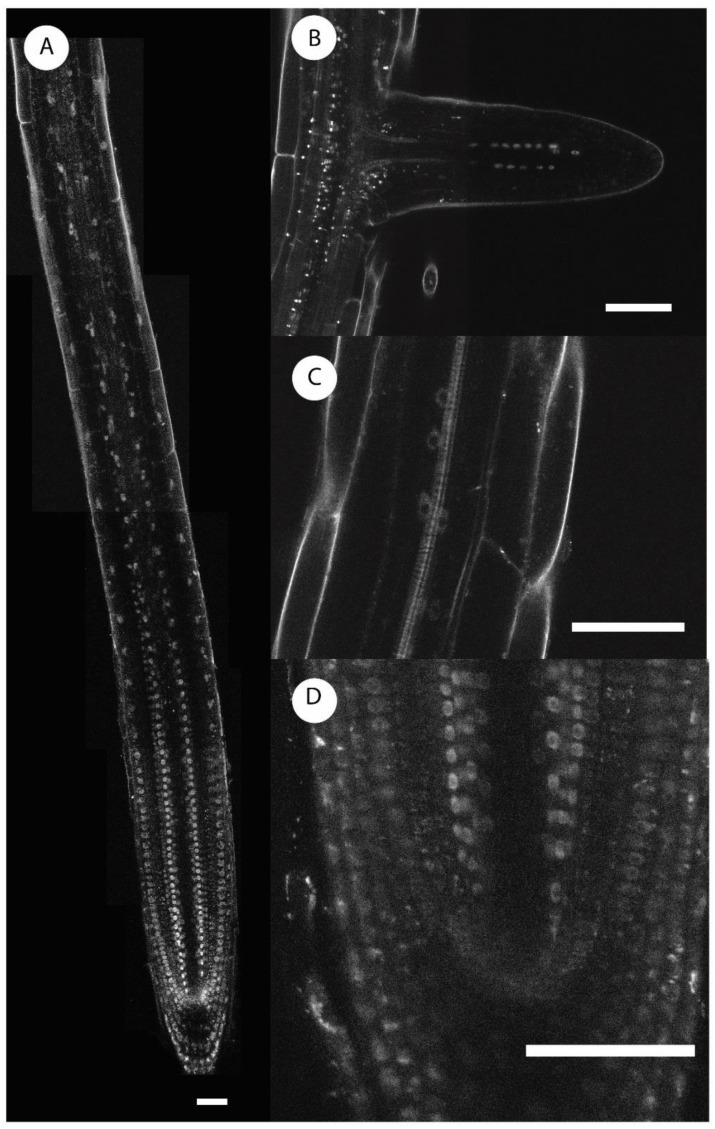
Localization of protein AHL18 in the primary root apical meristem and during lateral root primordium development. (**A**–**D**) *pAHL18::AHL18:mRuby*. Scale bars = 50 µm.

**Figure 3 ijms-21-01886-f003:**
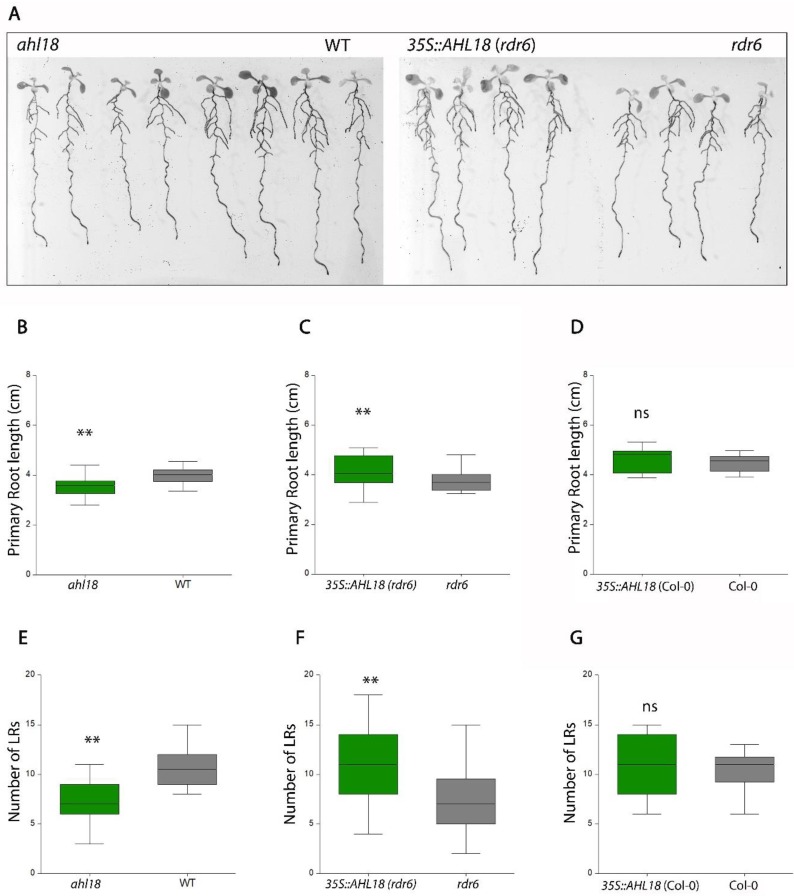
*AHL18* modulates root system architecture. (**A**) Overview of 7 DAG plants of *AHL18* loss of and gain of function lines on MS plates. (**B**–**D**) The length of primary root was shorter in the *ahl18* mutant compared with the overexpression line *35S::AHL18* in *rdr6* plants while *35S::AHL18* in Col-0 background remained without changes. (**E**–**G**) Number of lateral roots is reduced in *ahl18*, increased in *35S::AHL18* in *rdr6* and was without changes in *35S::AHL18* in Col-0. Statistically significant changes were detected at *p* ≤ 0.01 (**); Nested ANOVA, *n* = 64 (**B**,**E**), *n* = 48 (**C**,**F**). Data are consistent across three repeated experiments.

**Figure 4 ijms-21-01886-f004:**
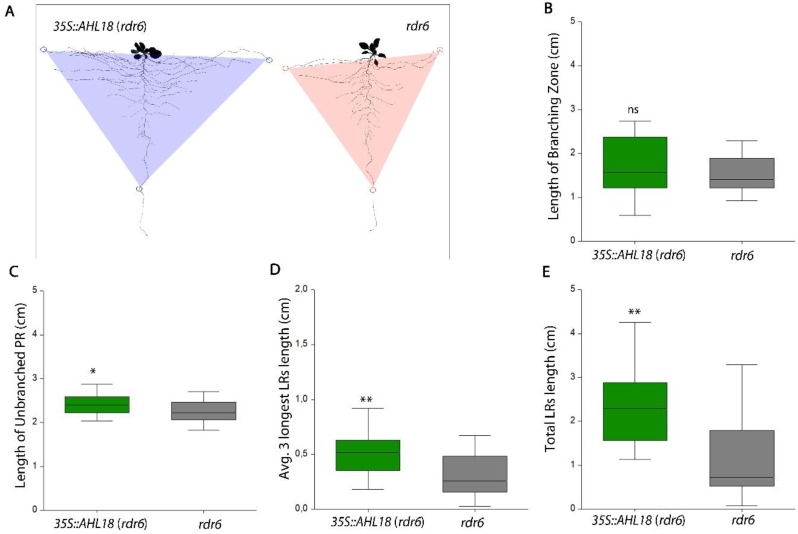
(**A**) Theoretical rhizosphere coverage in 14 DAG *35S::AHL18* and *rdr6* control plants. (**B**) Length of the primary root branching zone remains unchanged in overexpression of *AHL18*. (**C**) Length of unbranched zone of primary root as well as (**D**) average length of three longest lateral root and (**E**) total length of lateral roots in 7 DAG plants are greater compared to untransformed plants. Statistically significant changes were detected at *p* ≤ 0.05 (*); *p* ≤ 0.01 (**); Nested ANOVA, *n* = 48. Data are consistent across three independent experiments.

**Figure 5 ijms-21-01886-f005:**
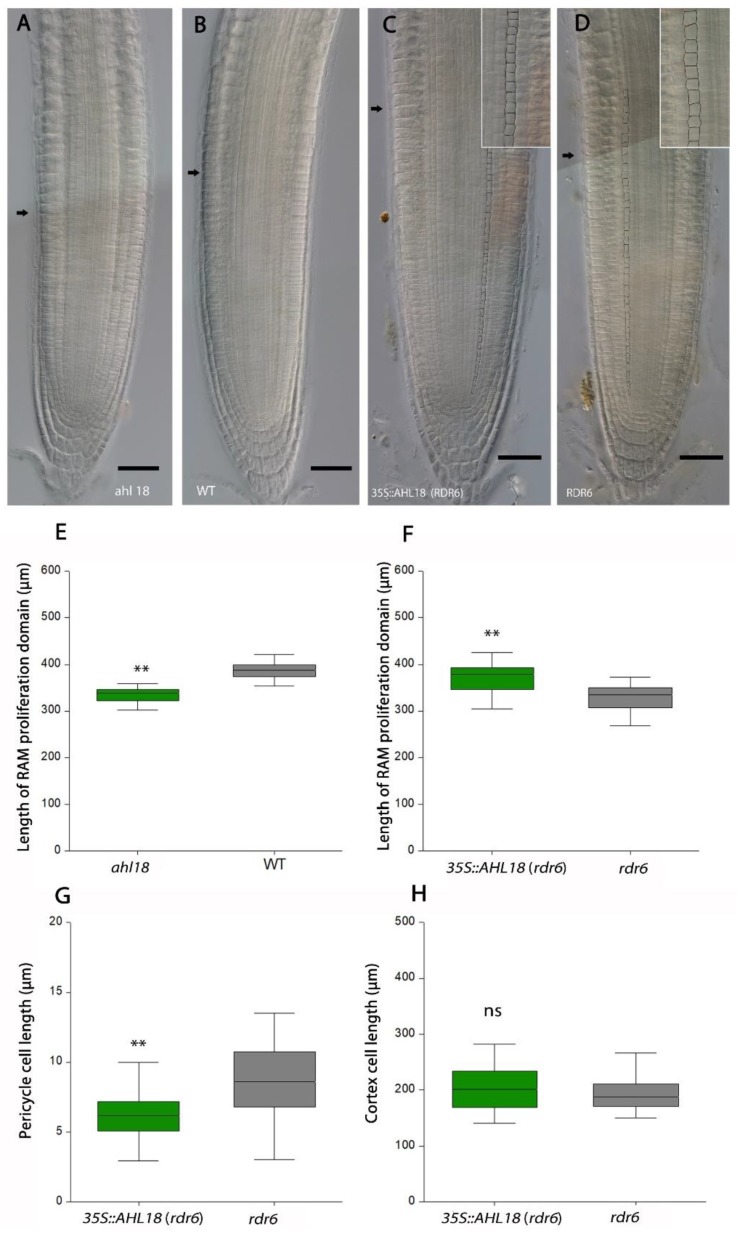
Length of the meristematic zone in (**A**) *ahl18* mutant and in (**C**) *35S::AHL18* line and respective control genetic backgrounds (**B**,**D**). Inlays demonstrates cell size of pericycle cells within the meristematic zone. Arrows show boundaries between proliferation and transition domains. (**E**) Length of RAM proliferation domain (µm) from 7 DAG plants displayed for the *ahl18* mutant and (**F**) *35S::AHL18* lines with respective controls. (**G**) Length of pericycle cells in RAM (µm) was shorter for *35S::AHL18* plants. (**H**) Length of fully elongated cortex cells measured at the level of the first emerged LR was unchanged compared to respective genetic background. Scale bars = 50 µm. Statistically significant changes were detected at *p* ≤.0.01 (**); Nested ANOVA, *n* = 54 (**E**), *n* = 38 (**F**). ANOVA, *n* = 16 (**G**,**H**). Data are consistent across two repeated experiments.

**Figure 6 ijms-21-01886-f006:**
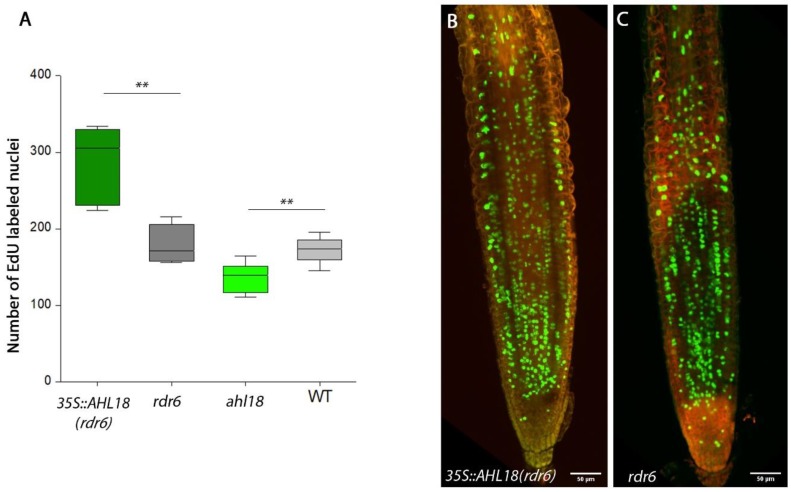
(**A**) Number of S-phase cell nuclei in the primary RAM labelled with EdU. (**B**,**C**) Distribution of dividing cells in meristematic zone of primary root in plants with an overexpression of *AHL18* and *rdr6* shown on 20 µm z-stacks projection. Scale bars = 50 µm. Statistically significant changes were detected at *p* ≤ 0.01 (**), (Two-sample t-test, *n* = 26).

**Figure 7 ijms-21-01886-f007:**
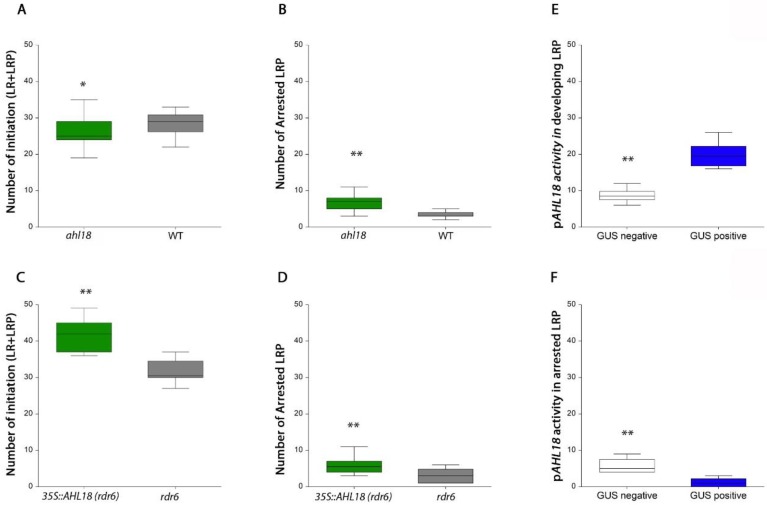
(**A**) Total number of lateral root primordia initiated in *ahl18* and (**C**) *35S::AHL18* and respective control plants. (**B**) Number of arrested or delayed root primordia for *ahl18* and (**D**) *35S::AHL18* detected in the branching zone. (**E**) Number of primordia with present or absent of *pAHL18::GUS* activity in developing primordia within the unbranched zone and (**F**) arrested or delayed root primordia in the branching zone in Col-0 background. Statistically significant changes were detected at *p* ≤ 0.05 (*); *p* ≤ 0.01 (**); nested ANOVA, *n* = 27 (**A**,**B**), *n* = 22 (**C**,**D**), *n* = 12 (**E**,**F**). Data are consistent across three independent experiments.

**Figure 8 ijms-21-01886-f008:**
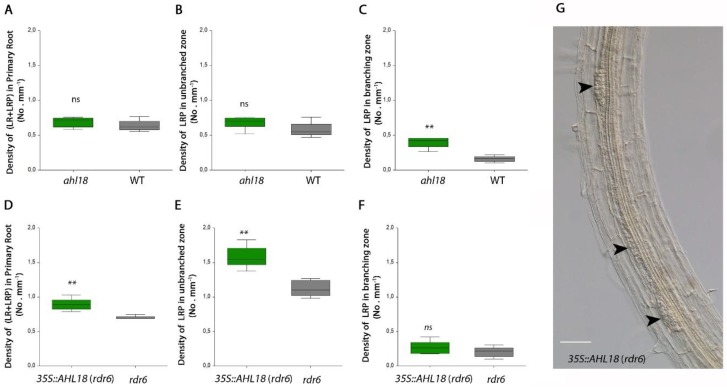
(**A**–**C**) Density of lateral root initiation in *ahl18* and (**D**–**F**) *35S::AHL18* and respective control plants according to the particular root zone. (**G**) Irregular spacing and higher density of LRP in *35S::AHL18* in *rdr6* plants background, arrowheads point to LRPs. Scale bar = 50 µm. Statistically significant changes were detected at *p* ≤ 0.01 (**); ANOVA, *n* = 11 (**A**–**C**), ANOVA, *n* = 12 (**D**–**F**).

**Figure 9 ijms-21-01886-f009:**
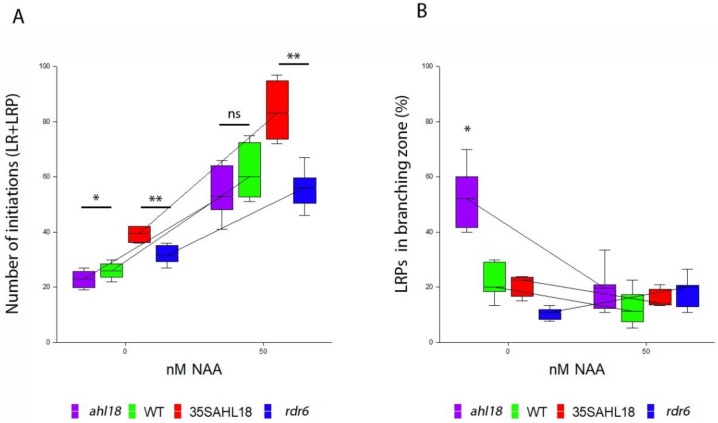
Effect of exogenous auxin on lateral root initiation in *AHL18* loss-of- and gain-of-function lines. (**A**) Higher LR initiation was observed across all lines with obvious and consistent trend. (**B**) Phenotype of restoration of arrested or delayed root primordia as percentage from total lateral root initiation events in branching zone was observed in *ahl18* plants when cultivated in medium supplemented with 50 nM NAA. Statistically significant changes were detected at *p* ≤ 0.05 (*); *p* ≤ 0.01 (**) between indicated pairs, ANOVA, *n* = 47 (**A**,**B**). Data are consistent across two repeated experiments.

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
