# Peer review of "At-Hook Motif Nuclear Localised Protein 18 as a Novel Modulator of Root System Architecture"

_ijms, 2020, doi:10.3390/ijms21051886_

Round 1
Reviewer 1 Report
In general, this manuscript is well-constructed and nicely organized.
The methodologies were appropriately adopted and described. The study also provided important information about the role of AHL18 in the formation of early root system architecture of A. thaliana alongside its expression domain and protein localization, pointing to its regulatory role in the apical meristem activity regulatory network and LRP development
I only have one minor concern.
The experiment numbers and samples should be added in the Material and Method section.
In figures 3, 4, 5, 7 y 8 the axis lables are no legible.
Please, revise the figure 9A. No statistically significant * changes were detected at the Figure A?
Line 254. Eliminate C letter
I suggest to authors including a Conclusion section. The manuscript do not objectively reflect the most significant findings.
Author Response
We really appreciate the comments and suggestion of the reviewer providing valuable feedback on our manuscript. We have adopted most of them as summarized below:
- The experiment numbers and samples should be added in the Material and Method section.
The information on number of replicates and independent experiments varies across experiments and used approaches. Placing of the same information into M&M section might be somehow duplicative and difficult to follow. Therefore we have placed only reference into M&M, where such information should be searched.
- In figures 3, 4, 5, 7 y 8 the axis lables are no legible.
Figures were modified to make the axis labels larger and easier to read.
- Please, revise the figure 9A. No statistically significant * changes were detected at the Figure A?
The Figure 9 was modified and pairs with statistically significant differences were marked.
- Line 254. Eliminate C letter
Done
- I suggest to authors including a Conclusion section. The manuscript do not objectively reflect the most significant findings.
We have included as a last paragraph of the discussion following text of conclusion: “AHL18 is as a novel regulatory element taking part in early stages of development of root system architecture. Its activity in the regulatory network controlling the root apical meristem proliferative activity, and onset of differentiation modulates growth of primary root. Experimental changes of AHL18 expression affects initiation of lateral root primordia and also their latter development. Increase in number of arrested/ developmentally delayed primordia among already emerged lateral roots, is one of traits associated with ahl18 phenotype. Presence of arrested primordia is well known phenomenon, which still awaits for clarification of the underlying regulatory network. We believe that deeper understanding of AHL18 regulatory function will extend our understanding of the processes mentioned above.”
Reviewer 2 Report
This should be in the attached Comments file

Author Response
We really appreciate the comments and suggestion of the reviewer providing valuable feedback on our manuscript. We have adopted most of them as summarized below:
- PG1: L75: Roles pointing to roles? AHL18 apparently is involved in regulation of activity of the apical meristem and in development of LRP (and a statement to that effect could be appropriate at the end of Introduction).
The last paragraph of the introduction was modified and we hope it better reflects those fundamental roles: “In the present study, we analyze the role of AHL18 in the formation of early root system architecture of A. thaliana alongside its expression domain and protein localization. There is clear regulatory function of AHl18 in the root apical meristem activity, onset of differentiation, LRP initiation and their latter development.”
- PG2: L146: Figures 4-9 could not be successfully converted by Acrobat to Word format that is necessary for appropriate editing with corrections and suggestions shown in sufficient contrast (font and color) and at precise locations. For the sake of final editing and type setting the authors need to recast the manuscript to an Acrobat version that can be properly transferred to Word format.
Unfortunately we are not able to solve mentioned format incompatibility, but MS Word version of the manuscript is uploaded for editorial usage.
- PG3: L148-152, legend of Fig. 4. Elucidation and interpretation of the effects should be in the main text, not in the legend. PG4: L163-170, legend of Fig. 5. Elucidations and interpretations need to be in the main text, not in the legend.
The minimal “interpretation” of data present in figure caption does not consume much of the space and in our opinion, the self-explanatory characteristics of the figures are enhanced in this way of presentation. That is why we prefer to keep the information.
- PG5: L197-198: It looks that 'was applied to allow for' should read 'was used toward a'.
Text was modified according to suggestion
- PG6: L224 (legend of Fig. 8): The first sentence is quite unclear and needs to be rephrased. Interpretations of the effects should be in the main text, not in the legend. PG7: L250-254 (legend of Fig. 9). In the legend it is necessary to state what precisely is shown in the graphs. Any interpretation of effects should be in the main text, with proper references to the graphs.
The legend of figures 8 and 9 was modified to improve the clarity. However, the minor “interpretation” of data present in figure caption does not consume much of the space and in our opinion, the self-explanatory captions of the figures are enhanced in this way of presentation. For this reason we prefer to keep present information.
- PG8: L263: Matsushita et al. [Ref. 25] deal with AGF1, which is not an AHL protein (but shares with AHLs the DNA-binding motif of AT hook type.) The 29 AHL variants are documented in Zhao et al. PNAS 110:E4688-97, and that reference should replace the current Ref. [14], also by Zhao (BMC Plant Biology 14, paper #266), which does not present the list of Arabidopsis AHL paralogues.
We really appreciate suggested modification, which indicates how knowledgeable reviewer about the topic is. The text was modified accordingly.
- PG9: L318: The authors need to avoid shop talk and shop condensations as much as possible; the text needs to be, as much as practical, intelligible to readers outside of plant protein genetics.
The text was modified to “initiating the cell elongation (i.e. the differentiation)”. We hope that this description will be generally understandable.
- PG11: L379: compensated?
Text was modified according to suggestion
- PG12: L385: Is there a mutation at position 469, or some interaction at that nucleotide?
The text was modified as follows to clarify meaning: “An A. thaliana (L.) Heynh. insertion mutant line for AHL18 gene (At3g60870, Sail_346_C06) with T-DNA flanking sequence insertion in exon at position 469 bp after the start codon, was obtained from publicly available collections...”
- PG13: L470: [Suggest change 'Beside all we thank' to something like 'We are especially thankful'; the 'Beside all' kind of implies a grudge that is nonetheless disregarded.]
Modified as follows: “We are especially thankful to Vojta Čermák and Adéla Přibylová for qPCR troubleshooting and whole Department of Experimental Plant Biology for friendly scientific environment.”
- PG14: L475: Abbreviations need to be aligned with the respective explanations
Formating was modified
- PG15: L509: here volume is 14, article number is 266; Zhao et al. PNAS 110:E4688 has the list of 29 Arabidopsis variants, and the BMC paper should be replaced by the PNAS paper
The reference was modified and BMC paper was replaced by the PNAS paper
Typos and stylistic changes indicated by reviewer directly in the text were mostly accepted and such changes are recorded directly in the MS Word text.